# Review of Cardiovascular Risk of Androgen Deprivation Therapy and the Influence of Race in Men with Prostate Cancer

**DOI:** 10.3390/cancers15082316

**Published:** 2023-04-15

**Authors:** James Fradin, Felix J. Kim, Grace L. Lu-Yao, Eugene Storozynsky, William K. Kelly

**Affiliations:** 1Sidney Kimmel Medical College, Thomas Jefferson University, Philadelphia, PA 19107, USA; 2Department of Pharmacology, Physiology, and Cancer Biology, Sidney Kimmel Cancer Center, Thomas Jefferson University, Philadelphia, PA 19107, USA; 3Jefferson College of Population Health, Thomas Jefferson University, Philadelphia, PA 19107, USA; 4Jefferson Heart Institute, Department of Medicine, Sidney Kimmel School of Medicine, Thomas Jefferson University, Philadelphia, PA 19107, USA; 5Department of Medical Oncology and Urology, Sidney Kimmel Cancer Center, Sidney Kimmel School of Medicine, Thomas Jefferson University, Philadelphia, PA 19107, USA

**Keywords:** androgen deprivation therapy, prostate cancer, cardiovascular disease, racial disparities

## Abstract

**Simple Summary:**

Prostate cancer is the second most deadly cancer in American men. The mainstay of treatment, androgen deprivation therapy, is aimed at modifying the hormonal signals involved in growth of prostate tumors. Recent research has revealed an association between these therapies and increased cardiovascular events like myocardial infarction and stroke. The goal of this review is to provide an improved framework for recognition of the cardiovascular risks of androgen deprivation therapy (ADT) and how to identify men who have an increased risk for a cardiovascular event while on ADT. Achieving this goal means physicians can more easily engage in shared decision making with patients and provide suggestions of cardiac optimization on an individual basis considering each person’s baseline risk factors. We also aim to initiate discussion surrounding the impact of racial disparities on the incidence and research of these cardiovascular complications.

**Abstract:**

Androgen deprivation therapy is the cornerstone of prostate cancer therapy. Recent studies have revealed an association between androgen deprivation therapy and cardiovascular adverse effects such as myocardial infarction and stroke. This review summarizes the available research on the cardiovascular risk of men using androgen deprivation therapy. We also discuss racial disparities surrounding both prostate cancer and cardiovascular disease, emphasizing the importance of biological/molecular and socioeconomic factors in assessing baseline risk in patients beginning androgen ablation. Based on the literature, we provide recommendations for monitoring patients who are at high risk for a cardiovascular adverse event while being treated on androgen deprivation therapy. This review aims to present the current research on androgen deprivation therapy and cardiovascular toxicity with an emphasis on racial disparities and provides a framework for clinicians to decrease the cardiovascular morbidity in men that are being treated with hormone therapy.

## 1. Introduction

Prostate cancer (PCa) is the second most common cancer in the United States and the second leading cause of cancer death in American men [1]. Localized and regional PCa have excellent 5-year survival rates approaching 100% [1]. Active surveillance has proven to be an effective long-term strategy since prostate cancer rarely metastasizes and causes death [2]. However, once prostate cancer has metastasized, androgen deprivation therapy (ADT) is the cornerstone of therapy, and ADT can be very effective, lowering serum testosterone to castrate levels within 2–4 weeks [3]. Despite its excellent outcomes, ADT has been associated with an increased risk of cardiovascular (CV) adverse events. Patients have been found to be at greater risk of myocardial infarction (MI), stroke, hypertension (HTN), and arrhythmia among other side effects. Cardiovascular disease (CVD) is the second leading cause of death in men with prostate cancer with two-thirds of patients at an increased risk for CVD [4]. The relationship between the increased CV risk of ADT and of PCa patients at baseline demands investigation. In addition, there is a need for clear management and surveillance guidelines for clinicians who have patients at high risk for CVD. This review revealed a lack of focus in recent literature on CV effects of ADT as it applies to different racial groups. Both CVD and PCa disproportionately affect different races, so it is necessary for future research to investigate the risk profile of ADT in the most at-risk sub-populations. This review aims to report up to date findings of the cardiovascular toxicity of ADT, provide preventative strategies for clinicians, and draw attention to racial disparities which need to be further evaluated in prospective clinical trials.

## 2. Materials and Methods

This literature review aims to investigate the effect of ADT on CV adverse effects and highlight the need for more investigation into the increased CV risk in patient sub-populations that may experience more risk at baseline. A review of current available literature was conducted specifically for ADT and CV risk. The reviewed literature was also assessed for data on CV risk in different races. PubMed and Cochrane databases were accessed. We performed a MeSH search of PubMed: ((“Prostatic Neoplasms/therapy” [Mesh]) AND (“adverse effects” [Subheading]) AND (“Cardiovascular Diseases” [Mesh])). We included clinical trial, meta-analyses, and randomized-controlled trials published in peer-reviewed scientific journals, in English with full-text availability. Inclusion criteria: publication after 2000, one or more of the following outcomes reported (adverse cardiovascular events, myocardial infarction, coronary artery disease, hypertension, cerebrovascular accident, arrhythmia). We summarize and discuss relevant results as well as provide suggestions for future research on this topic (Figure 1).

## 3. Androgen Deprivation Therapy: Overview and Mechanisms

PCa is a hormone-sensitive tumor that responds to changing levels of circulating androgens. A cornerstone of treatment for PCa is medical or surgical androgen deprivation therapy (ADT). ADT lowers androgen levels, inhibiting growth of tumor cells. The following section will elucidate the physiologic mechanism of ADT and discuss specific mechanism of action of available forms of ADT.

### 3.1. Rationale of Therapy

The rationale of therapy is that ADT reduces circulating androgen, preventing prostate tumor cells from using androgen as a growth factor for uncontrolled cell proliferation [5]. ADT targets the hypothalamic–pituitary–gonadal axis to exert its effects (Figure 2). The hypothalamus releases GnRH which binds GnRH receptors on the anterior pituitary, triggering release of luteinizing hormone (LH) and follicle-stimulating hormone (FSH). LH stimulates Leydig cells in the testes to release testosterone. Testosterone binds androgen receptors on tumor cells which results in translocation of the testosterone-receptor complex into the nucleus and activation of genes essential for cancer growth and proliferation. In the setting of PCa, this pathway has increased activity resulting in uncontrolled tumor growth.

### 3.2. Medical ADT

Medical ADT refers to the use of gonadotropin-releasing hormone (GnRH) agonists, GnRH antagonists, androgen receptor (AR) inhibitors, and cytochrome P450 17A1 (CYP17) inhibitors [6].

The mechanism of action of GnRH agonists relies on the negative feedback loop present in the hypothalamic-pituitary-gonadal axis. GnRH agonists initially cause a surge in FSH, LH, and testosterone. Continued elevated levels of GnRH will eventually lead to downregulation of the GnRH receptor on the anterior pituitary and, consequently, decrease release of LH and testosterone. GnRH antagonists cause immediate suppression of FSH, LH, and testosterone production. There is no hormone surge related to GnRH antagonists [6].

Additional therapy involves targeting the synthesis of androgens and the blockade of the androgen receptor. Androgen receptor inhibitors prevent testosterone from binding androgen receptors on prostate cells. First generation agents in this class include flutamide, bicalutamide, and nilutamide, which were commonly used in combination with GnRH agonists to mitigate the effects of the testosterone surge. Second generation androgen receptor inhibitors including enzalutamide, apalutamide, and darolutamide are more potent inhibitors which have shown improved overall outcomes in patients when combined with androgen deprivation therapy. These are the mainstay of therapy at this time [6].

Androgen synthesis inhibitors block the action of CYP17, an enzyme in the testes, adrenal glands, and prostate tumors which generates testosterone. Due to its effects on all androgen-producing tissue, androgen synthesis inhibitors reduce testosterone levels to a greater extent than any other treatment.

### 3.3. Surgical ADT

Surgical ADT classically refers to bilateral orchiectomy. By removing the testes, testosterone production is significantly reduced, depriving prostate tumor cells of their growth signals. Surgery results in permanent suppression of testosterone levels in the patient.

## 4. Cardiovascular Risk of Hormone Therapy

Androgen deprivation therapy, as a hormone-based treatment, comes with a few side effects. Some of the most common include decreased bone density and fracture [7], erectile dysfunction [8], decreased libido, hot flashes [9], metabolic effects (weight gain, increased body fat, insulin resistance), and CV effects.

Recent research has focused on the increased risk of CV events associated with ADT. Numerous studies have found increased incidence of myocardial infarction (MI), stroke, arrhythmia, hypertension, and sudden cardiac death (SCD) in men receiving ADT [4,5,10]. Cardiovascular disease is already the second leading cause of death in men with prostate cancer. It is estimated that two-thirds of men with PCa are at increased risk for CVD [4]. It is crucial to mitigate increased CV risk in a patient population already at high-risk for CVD [10]. The following sections elaborate on mechanisms underlying the increased CV risk, metabolic changes that patients may experience, preclinical models of ADT-induced CVD, and current ADT clinical trial data.

### 4.1. Mechanisms of CV Risk and Metabolic Considerations

The mechanisms for the increased CVD associated with ADT are not completely elucidated. ADT has been shown to lead to the development of metabolic syndrome, a collection of risk factors for cardiovascular disease and type II diabetes mellitus. Characteristics include insulin resistance, glucose intolerance, hyperinsulinemia, increased very-low-density lipoprotein (VLDL), hypertriglyceridemia, decreased high-density lipoprotein (HDL), and hypertension. Diagnostic criteria for the syndrome include 3/5 of the following criteria: fasting plasma glucose level > 110 mg/dL, serum triglyceride level ≥ 150 mg/dL, serum HDL < 40 mg/dL, waist circumference > 102 cm, and blood pressure ≥ 130/85 mmHg [11]. A study of Asian men with prostate cancer who received ADT showed significant changes in metabolic parameters during the treatment course [12]. A 4.8% increase in fasting blood glucose and 2.7% increase in HbA1c were both noted at 6 months as well as increased triglycerides and mean body weight during follow-up. This study, among many others, has shown a clear association between ADT and metabolic changes that could potentially predispose patients to CVD (e.g., hyperlipidemia, elevated fasting blood glucose, hypertriglyceridemia, and increased weight) [13,14]. Other studies have shown that men treated with ADT develop increased fat mass located in subcutaneous tissue, decreased muscle strength, decreased sensitivity to insulin, and higher fasting insulin levels [10]. Other studies have suggested that changing hormone levels influence endothelial cell stability and activation, leading to conditions that favor formation and destabilization of atherosclerotic plaques [15,16]. Destabilization of plaques, visceral adiposity, insulin resistance, and endothelial dysfunction result in overall increased risk of MI, stroke and hypertension [6]. However, the exact mechanisms of ADT-induced CVD remain unclear and require more research.

While we still have not found the reasons for why these metabolic changes exist, it is important to recognize them in the context of the PCa patient population which frequently suffers from metabolic conditions prior to starting therapy. Many of these conditions have been found to bed directly associated with worse outcomes. Obesity, measured by body mass index (BMI) and waist circumference, has been found to be associated with higher mortality and higher-grade disease in PCa [17,18]. Data are lacking on whether weight loss could serve a preventative role in PCa patients, and more research into the mechanisms underlying these weight changes is needed to back clinical decision-making. Metabolic syndrome, discussed earlier in this review, has been found to be associated with a higher risk of fatal prostate cancer as well [19]. Hyperlipidemia is important to consider as cholesterol is a substrate for androgen synthesis. Studies have shown an association between triglycerides levels and recurrence rate of prostate cancer [20]. It is important for clinicians to consider the impact of these metabolic abnormalities on the course of their patients’ tumors and to incorporate the changes expected with hormonal therapies into proper monitoring.

### 4.2. Preclinical Models

Murine models have been used to investigate the association of ADT with cardiovascular events as well as the mechanisms underlying the observed increased risk. An investigation of GnRH agonist leuprolide in adult mice found that mice receiving leuprolide experienced significant increase in abdominal weight without a change in total body weight, increased cardiac troponins, but no echocardiographic findings of systolic or diastolic dysfunction [21]. Another study of ApoE^−/−^ mice with established atherosclerotic plaques compared those receiving leuprolide (GnRH agonist) vs. degarelix (GnRH antagonist) [22]. Mice receiving leuprolide were found to have more areas of necrosis and inflammation within plaques reflecting an increased risk of plaque rupture. Another study of ApoE^−/−^ mice found leuprolide again caused increased atherosclerosis in orchiectomized mice compared to similar mice receiving degarelix [23]. Interestingly, in the same study, there was no difference in atherosclerosis in mice who remained intact. These preclinical models support some of the proposed mechanisms of the increased CV risk of ADT like increased adiposity and increased atherosclerosis as well as raise interesting questions about the underlying mechanisms leading to the observed physiologic changes.

Whether or not ADT is the cause of the increased incidence of CVD is still a point of contention. Observational studies and random clinical trials have been conducted that found a significant positive association between ADT and certain CV events while others have found no significant association (see Appendix A). More recent research focuses on CVD outcomes of each class of ADT (i.e., GnRH agonists vs. antagonists vs. orchiectomy) and outcomes of patients with and without previous CVD (see Appendix A).

### 4.3. Evidence of CV Risk of ADT

In 2006, Keating et al. were the first to publish data regarding an increased incidence of CVD and diabetes in men receiving ADT [24]. They conducted an observational study of 73,196 Medicare enrollees, aged 66 years or older, diagnosed with prostate cancer between 1992 and 1999. The study found GnRH agonists to be associated with an increased incidence of diabetes (HR, 1.44; *p* < 0.001), coronary heart disease (HR, 1.16; *p* < 0.001), myocardial infarction (HR, 1.11; *p* = 0.03), and sudden cardiac death (HR, 1.16; *p* = 0.004). These results demonstrated a need to further characterize the nature of the relationship between ADT and CVD as well as identify patients at high risk for these adverse events and develop strategies of prevention. Since 2006, many other studies have supported these initial findings. A 2015 study revealed a similarly increased risk of CVD in men receiving a GnRH agonist for treatment of prostate cancer (HR, 1.21; 95% CI, 1.18 to 1.25) [25]. Jespersen et al. conducted a national cohort study using the Danish Cancer Registry investigating the incidence of MI and stroke in men receiving ADT. The study found a significantly increased risk of MI (HR, 1.31; 95% CI, 1.16–1.49) and stroke (HR, 1.19; 95% CI, 1.06–1.35) in men receiving ADT [26]. Furthermore, men with no preexisting MI or stroke had a significant increase in risk of MI or stroke on ADT, while men with one or more prior MI or stroke showed no significant increase in risk of additional CV events. A 2013 observational study of 185,106 US men with prostate cancer receiving ADT revealed an increased risk of MI (HR, 1.09; 95% CI, 1.02–1.16) and diabetes (HR, 1.33; 95% CI, 1.27–1.39) in patients with no comorbidities [27]. The study also revealed that men with significant comorbidities including prior MI, stroke, hypertension, and COPD had an increased but non-significant increase in MI and stroke while receiving ADT. Not all studies agree on the increased risk of cardiovascular events in men receiving ADT. Efstathiou et al. conducted a large phase III randomized trial of 945 men treated with or without goserelin (GnRH agonist) after radiation therapy for locally advanced prostate cancer [28]. They found that the treatment arm had no significant change in cardiovascular mortality as compared to the control group receiving just radiation therapy (HR, 0.73; 95% CI, 0.47–1.15; *p* = 0.16). Of note, this was one of the few trials that compared the treatment group receiving ADT to a control group receiving radiation as opposed to a different ADT drug.

### 4.4. Types of Medical ADT and CVD Risk

Studies have also focused on the risk of cardiac events in men with preexisting CVD as well as outcomes in those treated with GnRH antagonists as compared to GnRH agonists. One study pooled data from three prospective randomized trials and found that men with preexisting CVD had significantly decreased risk of cardiac events within 1 year of beginning treatment when on a GnRH antagonist as compared to a GnRH agonist (HR, 0.44; 95% CI, 0.26–0.74; *p* = 0.002) [29]. Another study comparing GnRH agonists and antagonists in men with prior CVD found a decrease in risk of heart failure (HR, 0.46; 95% CI, 0.26–0.79) and ischemic heart disease (HR, 0.26; 95% CI, 0.11–0.65) [30]. A 2019 phase II randomized trial compared the cardiovascular risk of GnRH agonists and antagonists [31]. This study corroborated previous research, finding a significant reduction in major cardiovascular and cerebrovascular events after one month of ADT using GnRH antagonists as compared to agonists (ARR, 18.1%; 95% CI 4.6–31.2, *p* = 0.032). Another systematic review compared available data on CV risk between GnRH agonists and degarelix. This study found degarelix to be protective for CV risk, to reduce new CV events, and to reduce CV interventions [32]. Relugolix, another new GnRH antagonist, was found by a recent phase III trial to have improved testosterone suppression as compared to degarelix along with a 54% reduction in major adverse cardiovascular events [33].

Other studies focused on the CV risk with androgen receptor inhibitors and androgen synthesis inhibitors. Recent meta-analyses of the use of abiraterone and enzalutamide in men with PCa revealed significant increases in adverse cardiac events and HTN [34,35]. A 2020 study revealed that men with a preexisting history of three or more CVDs who were taking abiraterone or enzalutamide experienced a significant increase in 6-month mortality as compared to men with no prior CVD (RR, 1.56; 95% CI, 1.29–1.88) [36]. Another meta-analysis revealed an increased risk of CV adverse events (RR, 1.41; 95% CI, 1.21–1.64) and HTN (RR, 1.79; 95% CI, 1.45–2.21) in men receiving abiraterone for prostate cancer treatment [37].

An important point to note is that men with clinically apparent CVD represent a significant population to monitor for complications while receiving ADT, but men with sub-clinical CVD must be considered as well. These patients could be at increased risk for complications while on hormonal therapy due to lack of regular monitoring for conditions like hypertension, diabetes, and coronary artery disease, among others. Additionally, patients with uncontrolled hypertension or significant preexisting CVDs are often excluded from clinical trials; therefore, the findings from the clinical trials might not be applicable to patients with significant preexisting CVD. Additionally, many trials have limited follow-up and sub-optimal data on participants’ previous CVD and risk factors. Larger-scale trials and longer follow-up with more in-depth patient demographics and risk factors will be needed to fully understand the present risk of ADT [32]. As of now, official guidelines have only come from the FDA, who in October 2010, determined that the available evidence showed a small but significant increase in risk of CV events and mandated labels on GnRH agonists to show safety information regarding its CV risks [38].

## 5. Racial Disparities

The racial disparities of cardiovascular disease and prostate cancer are striking. Both diseases affect men of African descent at a younger age, and outcomes are poorer in this demographic than in any other [39]. Genetic, environmental, behavioral, and socioeconomic factors explain why African American men experience a greater burden of disease from CVD and PCa. However, prospective studies are lacking on the impact of race on the overall cardiovascular risk in men treated with ADT.

### 5.1. Race and CV Health

Coronary heart disease (CHD) is responsible for an estimated one in seven deaths in the USA [40]. Risk factors for CHD and other CVD include hypertension, obesity, diabetes, high cholesterol, smoking, alcohol, reduced physical activity, age, and race among many other factors [41]. In addition to race by itself being a risk factor for CVD, recent data from the CDC reported that non-Hispanic black populations were the group most likely to have HTN, obesity, and diabetes regardless of genetic risk factors. In 2017, 9.5% of non-Hispanic black people 18 years and older had CHD [41]. African Americans specifically experience a significantly increased burden of disease as compared to white American populations. They have been found to suffer more from coronary heart disease, heart failure, sudden cardiac death, and cerebrovascular disease/stroke than white Americans of the same age and gender [39,40,42]. Rates for African Americans are 20% higher for heart disease and 40% higher for stroke compared with rates in white Americans [43].

Social determinants of health are a major factor contributing to the increased burden of CVD in minority populations. These determinants include economic stability, access to and quality of education, access to and quality of healthcare, neighborhood environment, and social/community environment [44]. Social determinants of health have potential to drive health inequities in communities of color, predisposing those populations to a greater risk of poor health outcomes [44]. For example, African Americans with confirmed acute coronary syndrome (ACS) were younger, poorer, and less educated and had a longer pre-hospital delay than white Americans [40]. Another reason for increased prevalence of CVD in minority populations is a lack of health insurance. Ethnic and racial minorities constituted 55% of the uninsured population prior to the passage of the Affordable Care Act, and people with less insurance tend to engage less with the healthcare system for monitoring of risk factors like cholesterol and blood pressure [40].

The hereditary component of race has been identified as an independent predictor of survival [45]. Ischemic arterial disease is the result of an occlusive thrombus formed by a ruptured atherosclerotic plaque. Thrombus formation is promoted by different substrates and signaling pathways including platelet activation. A 2013 study by Edelstein et al. investigated thrombin receptors PAR1 and PAR4 which mediate thrombin signaling leading to platelet activation. They found that black research subjects had significantly increased incidence of a PAR4 variant associated with increased platelet aggregation [45,46]. This mechanism represents one pathway of platelet activation, but a definitive mechanism is still unknown. Platelet reactivity is known to be heritable, so it represents one way that genetic differences can predispose certain populations to increased risk of atherosclerotic disease. An additional component of increased risk could be structural heart disease. A multicohort study found an association between Bcl2-associated anthanogene 3 (*BAG3*) and negative outcome in cases of dilated cardiomyopathy in African Americans [47]. The study found four *BAG3* variants which were not present in a reference population of European ancestry that were associated with a significantly increased hazard ratio in African Americans with dilated cardiomyopathy (HR, 1.97; 95% CI, 1.19–3.24). The findings of these studies involving platelets and *BAG3* variants suggest a molecular component to race-related increases in CV disease and may represent the future of risk-assessment.

Another population of interest is Hispanic/Latino men with prostate cancer. A 2007 study looking at cardiovascular morbidity associated with ADT found that Hispanic/Latino men had a lower CV morbidity than other races [48]. The authors of this study mention a previously documented “Hispanic paradox” in which these populations experience lower cardiovascular mortality despite increased risk factors [49]. Another major demographic to consider is Asian populations. There have been quite a few publications describing the CV adverse events of ADT in Asian populations. The results of these studies are like those discussed earlier in this review. Some have found a significant occurrence of adverse events like acute MI, stroke, and increased cardiovascular mortality. while others have found no association [50]. An increased risk of suffering from CV adverse events while on ADT has been proposed by a couple of mechanisms. Asians have been found to have the longest CAG repeats out of any ethnicity, and these repeats may be associated with increased or decreased insulin sensitivity in response to testosterone levels [51]. These sources of decreased and increased risk of adverse cardiac events in Hispanic/Latino and Asian people, while still requiring more investigation, may represent risk factors that future clinicians will have to consider when starting their patients on ADT.

### 5.2. Race and PCa

Prostate cancer does not affect all people equally. African American men specifically experience an increased burden of disease from PCa. They are 50% more likely to develop prostate cancer in their lifetimes compared with white men, and they die because of prostate cancer at a rate greater than twice that of white men [52,53]. Five-year survival for all stages is higher for white men than black or Hispanic men [54]. There are quite a few proposed explanations for this disparity including genetics and heritability, differences in screening and detection, and socioeconomic status along with access to healthcare.

There is significant genetic variation in the mutations and heritability of PCa in African American men as compared to European-American men [55]. Various genetic loci have been identified that are more common in populations of African descent that confer increased risk of PCa development and progression. One example is the BMP2 (20p12) and CXCR4 (2q22) genes, which are both associated with bone metastases and enhanced PCa adhesion, migration, and invasion. There is disproportionate upregulation of these genes in African American men, potentially explaining increased rates of early metastasis [56]. Recent research has found that use of genomic classifiers like the Decipher system (a 22 gene panel that predicts risk of metastasis) can help identify African American men with aggressive disease who may benefit from earlier intervention and specific therapy [57].

Prostate-specific antigens (PSA) are popular for screening of PCa. While screening has been found to have equal benefits among different races, studies have shown African American men have higher PSA values at the time of diagnosis [55]. This finding could suggest accelerated disease progression in African American males but also could be a manifestation of poor access to health resources. Other relevant forms of detection of PCa include the digital rectal exam (DRE) and biopsy. DRE and biopsy have traditionally underperformed in detecting anteriorly located tumors which have a higher incidence in African American men [55].

Another factor which cannot be ignored is socioeconomic status and access to healthcare. In a 2013 trial, Barocas et al. determined that African American males aged less than 65 years were 45% less likely to undergo a diagnostic evaluation for PCa following positive screening compared to European-American men [58]. One proposed reason for this difference is lack of access to healthcare. Other factors include mistrust in the healthcare system among minority populations and feelings of cancer fatalism, that one will ultimately die from cancer no matter what treatment they receive [59]. Regardless of the reason, minorities receive less diagnostic testing which could otherwise lead to earlier interventions and treatment. Access to medical resources also adds to racial disparities in treatment outcomes. Non-white men treated in high-technology, high volume medical centers received more management for PCa relative to those who were treated in lower technology locations [60]. Minority populations, unfortunately, experience unequal access to these resources and thus experience inferior treatment for PCa as compared to those with access to more capable medical centers. Less screening, less access to modern healthcare resources, and development of disease at a younger age all lead to poorer outcomes for African American men with PCa.

### 5.3. Challenges Ahead

Through our review of the literature, we found a paucity of data investigating race-specific outcomes related to ADT. Currently, we only have indirect evidence of the increased CV risk in minority populations receiving ADT. Most clinical trial data come from studies with low numbers of minority participants and were not structured to collect race-specific data, thus making it difficult for us to determine risk based on race. In addition, most studies only use self-identified race data and not genomic based data, leading to potential bias. As we have already discussed, the African American population along with other minority populations experiences an increased risk of death due to CVD and PCa. The findings of our review motivate us to call for clinical trials which are designed to collect this data with a goal of targeting more at-risk populations. There are already trials that provide a model for risk-stratification of African American men with PCa. One recent study investigated subsets of African American men with localized PCa, aiming to use genomic methods to risk-classify patients [57]. Studies like this have potential to allow providers to choose appropriate screening and preventative measures for their patients in an individualized manner.

## 6. Risk Management

The increased CV risk of ADT demands a discussion between the patient and physician. There will be circumstances in which high-risk patients choose to begin hormonal therapy despite its risks. Appropriate efforts must be made by the medical team to monitor and reduce the occurrence of cardiovascular events. We suggest a further modified version of the “ABCDE” algorithm (Figure 3) published by Campbell et al. to manage cardiovascular risk in cancer patients [61,62]. Our model includes “diversity” under the “D” section to emphasize differences in risk among different racial backgrounds. This strategy would be an appropriate framework, and we suggest patients should follow up with their physicians every 3 months during treatment. Patients at average risk may experience serious adverse effects from ADT as well, so they must receive proper monitoring. Physicians should engage in discussion surrounding the importance of regular monitoring of blood pressure and blood lipids to recognize early changes which may increase the risk of CVD.

### 6.1. ABCDE Algorithm

#### 6.1.1. A: Awareness, Aspirin, and Arrhythmia

Patients receiving ADT should be aware of the signs and symptoms of CVD and seek appropriate medical attention. They may also benefit from counseling on metabolic effects like weight gain and increased waist circumference. Weight and waist circumference can be measured at regular follow-up appointments. Aspirin can be considered for primary and secondary prevention of CV adverse events. Data are conflicting regarding the effects of aspirin on prostate cancer-specific mortality (PCSM). One study found aspirin to be associated with lower PCSM in high-risk prostate cancer patients (HR, 0.60; 95% CI, 0.37–0.97) [63]; however, a recent meta-analysis found no statistical difference in PCSM for men taking aspirin regardless of risk level [64]. There is research showing aspirin clearly benefits patients in reducing major cardiac adverse events (MI, stroke, etc.) (RR 0.89, 95% CI 0.85–0.94, *p* < 0.001) [65]. The preventative effects of aspirin have not been studied in patients receiving ADT and this could represent an area of future research in reducing cardiac complications. We recommend aspirin as preventative therapy for patients with known coronary artery disease, peripheral arterial disease, and cerebrovascular disease receiving ADT. Patients should be vigilant for signs of arrhythmias (i.e., palpitations, syncope) as they are associated with some hormonal therapies.

#### 6.1.2. B: Blood Pressure and Biomarkers

Patients should maintain blood pressure below 140/90 mm Hg. Increased blood pressure is a well-known risk factor for the development of atherosclerosis. Although there are no specific biomarkers associated with increased risk of CVD and PCa treatment, biomarkers like BNP (Brain Natriuretic Peptide) and troponin may be helpful to monitor to assess the risk of cardiovascular events [66].

#### 6.1.3. C: Cigarettes, Cholesterol, and CT/Cardiac Imaging

Smoking should be ceased as it has been found to be an independent risk factor for PCa [67]. Hyperlipidemia should be managed with appropriate guideline-based therapy. Baseline lipid profiles should be collected before initiating treatment in order to monitor for new hyperlipidemia or worsening of preexisting disease. Cardiac CT may be used for detection of atherosclerosis. Echocardiogram can detect more serious cardiomyopathies that may increase risk of cardiac events while on ADT.

#### 6.1.4. D: Diabetes, Diet, Diversity, and Drug Choice

As mentioned earlier, ADT can have significant effects on blood glucose. Diabetic patients receiving ADT should frequently monitor blood glucose, and diabetes therapy should be adjusted as needed. Patients without diabetes should have their blood glucose monitored since ADT can lead to the development of diabetes. Patients should consume a heart-healthy diet to further minimize cardiovascular risk factors. The diversity term encourages clinicians to identify those who may be at increased risk of CV complications due to racial or ethnic background. Identification can be accomplished with thorough family and social history. Although more research is required, biological markers of increased CV risk represent a promising method of recognizing high-risk patients in the future. As referenced earlier, this can involve assessment of platelet dynamics or presence of certain genetic variants that segregate in racial groups. Since ADT drugs have different CV risk profiles, oncologists, cardiologists, and patients should engage in shared decision making to find the best therapy to minimize risk.

#### 6.1.5. E: Exercise, EKG, and Exposure

Exercise is essential to reduce the risk of cardiovascular complications. Patients should remain active and engage in a variety of exercise intensities. As mentioned earlier, arrhythmias may develop with certain hormonal therapy. Regular EKGs may be used to monitor for development of abnormal heart rhythms or electrical abnormalities. Finally, length of hormone therapy must be considered as longer duration may increase risk of adverse events.

**Figure 3 cancers-15-02316-f003:**
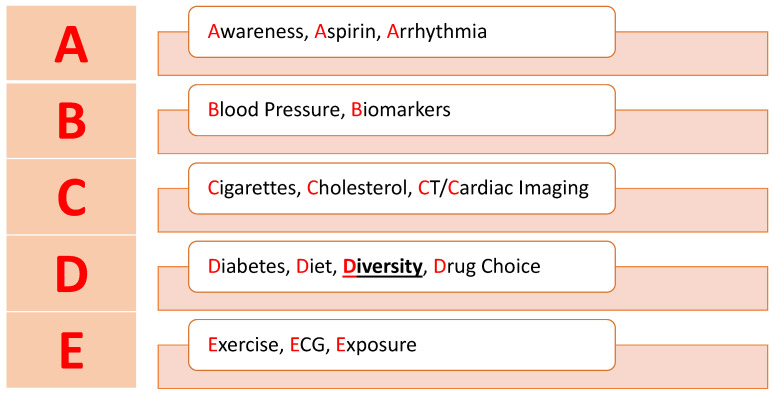
Modified ABCDE algorithm. See newly added “Diversity” under the “D” section.

### 6.2. Authors’ Suggestions for ADT Choice in High-Risk Patients

Following current meta-analysis and large scale randomized controlled trial outcomes, there are several considerations regarding the most appropriate ADT for patients at high-risk for cardiovascular complications. First, data suggest the use of GnRH antagonists over agonists. As detailed earlier in this paper, many researchers have found that GnRH antagonists like degarelix have fewer associated CV adverse events than their agonist counterparts [24,26,29,31]. Relugolix may be given additional consideration due to recent clinical trial data showing very low associated CV risk [68,69]. Androgen synthesis inhibitors (i.e., abiraterone) and androgen receptor antagonists (i.e., enzalutamide) have become key agents for castration sensitive and resistant PCa; however, they also can carry increased CV risk, and clinicians should take appropriate measures to monitor and modify their CV risk factors. Newer agents like darolutamide and apalutamide have potential to be effective treatment options with lower risk, but more trial data are required [70].

## 7. Conclusions

In conclusion, prostate cancer patients are at high risk for cardiovascular complications. The cardiovascular toxicity of androgen deprivation therapy poses an additional threat to their health. While this association has been reproduced in many studies, the relationship between prostate cancer hormonal therapy and cardiovascular toxicity requires more research. There is a need to perform more prospective clinical studies to clearly define the increased risk that patients may experience with these agents. In addition, further research must address populations that experience marked increases in cardiovascular morbidity. Men of African descent experience an increased burden of CVD and PCa independent of the threat of ADT. This patient population must be considered in research due to this significantly increased risk. Those most at risk should be represented and investigated as part of the studies whose findings will ultimately guide their treatment. This racial disparity may be caused by determinants of health like decreased access to healthcare, advanced facilities, and proper nutrition. This increased risk may be due to genetics and inherited predisposition to cardiovascular disease from molecular mechanisms like platelet dynamics and genetic variants. We believe that the future of assessing individual CV risk involves detecting these molecular anomalies and that researchers should continue to investigate a molecular mechanism of risk stratification.

## Figures and Tables

**Figure 1 cancers-15-02316-f001:**
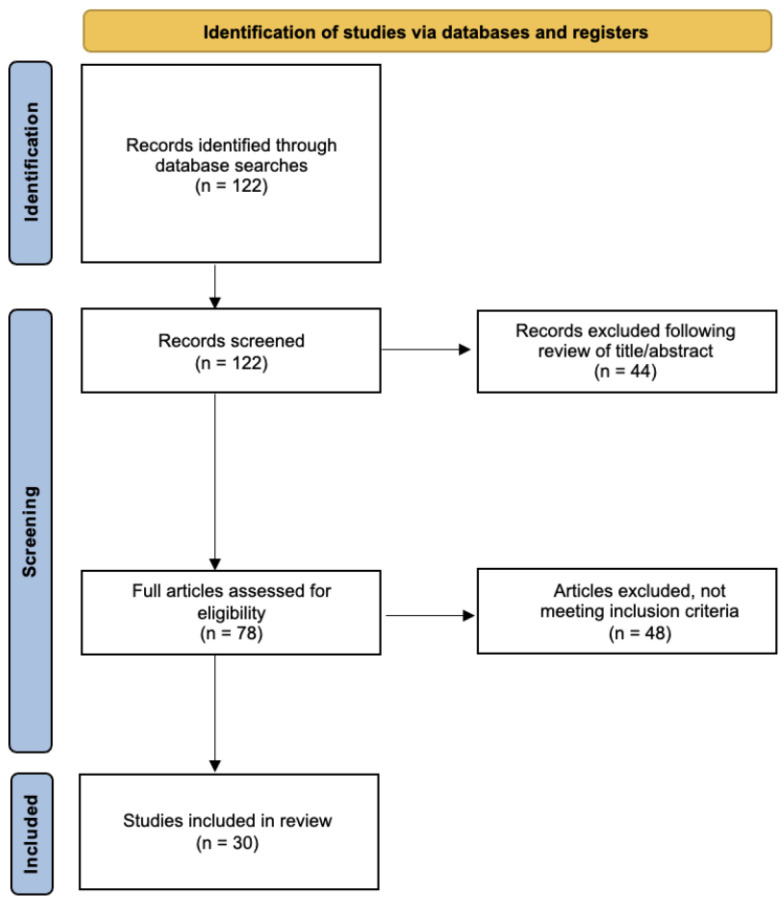
Results of the literature screening process.

**Figure 2 cancers-15-02316-f002:**
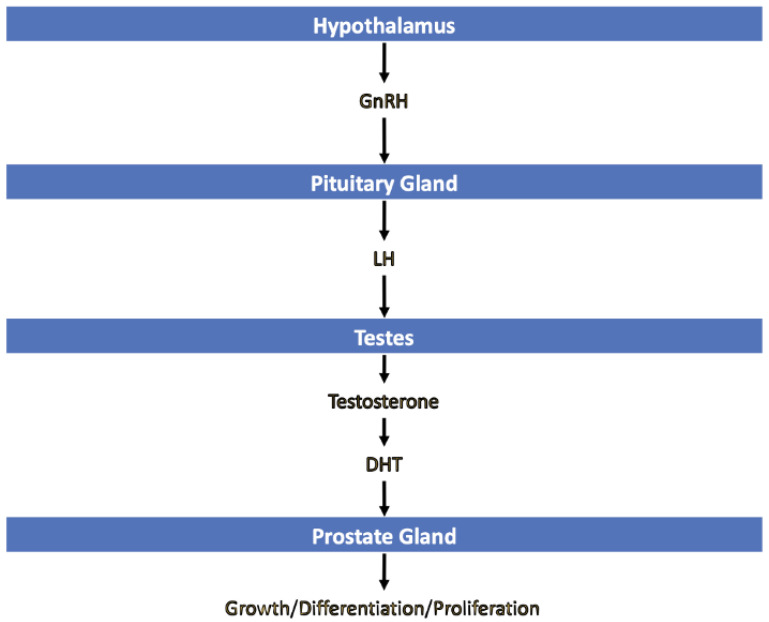
The hypothalamic–pituitary–gonadal axis control of the prostate.

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
