# Peer review of "Review of Cardiovascular Risk of Androgen Deprivation Therapy and the Influence of Race in Men with Prostate Cancer"

_cancers, 2023, doi:10.3390/cancers15082316_

Round 1

Reviewer 1 Report (Previous Reviewer 1)

I have not further suggestions/comments, since Authors have addressed all of them in this improved version of the manuscript

Author Response

Thank you for your time and prior suggestions for this manuscript.

Reviewer 2 Report (Previous Reviewer 3)

The revised manuscript by Fradin et.al. included new literature on the cardivascular effects of ADT for the treatment of prostate cancer. In addition, authors also included a short discussion on the challenges on this research topic. The current version is much improved and reflected the critical evaluation of literature by authors. There are no additional comments at this time. 

Author Response

Thank you for your time and prior suggestions for this manuscript.

Reviewer 3 Report (New Reviewer)

The article by Fradin et al. describes a compilation of literature on ADT associated with cardiovascular events. The reviewed manuscript is fascinating in a few aspects, such as the correlation of CVD and PCa in racial disparity. However, recently various review articles compiled similar aspects, which end of not much new information in this aspect.

Neither the review gives mechanistic evidence of ADT-induced CVD nor incorporates the studies done in the preclinical model.

The review article misses much critical information

1.       The term ADT is broad in terms of the target in PCa patients. Various review articles captured maximum difference between the different ADT agents and CVD – These need to be discussed in detail

2.     Similarly, PCa is an age-related disease and always co-occurs with metabolic disorders. A section on Metabolic risk factors is necessary to discuss the ABCADE algorithm and CVD risk in race

3.     Since mechanistic studies are not possible in human tissue, the authors can discuss the results from the preclinical model (PMID: 36878484; PMID: 27189011, PMID: 26908406).

Minor comments

1.       Line 142-143 is not necessary

2.       The section of Race and PCa can be reduced as the focus is not on prostate cancer.

3.       The table describes abiraterone acetate and enzalutamide should be combined since there was always overlap between the treatments and patient population.

Author Response

The authors once again thank the reviewers and editors for their continued consideration of our manuscript. See below our revisions in response to the reviewers’ suggestions.

  1. The term ADT is broad in terms of the target in PCa patients. Various review articles captured maximum difference between the different ADT agents and CVD – These need to be discussed in detail

Additional data was included on the difference between ADT agents and CVD. Under the section “Types of Medical ADT and CVD Risk”. These additions supplement the existing discussion on ADT agents and their CVD impact which already discusses in detail the effects of ADT in all forms, GnRH agonists vs. antagonists and, androgen receptor inhibitors, androgen synthesis inhibitors. This is also in addition to the data found in the appendix which details more of the cardiovascular risk profile of these agents.

  1. Similarly, PCa is an age-related disease and always co-occurs with metabolic disorders. A section on Metabolic risk factors is necessary to discuss the ABCADE algorithm and CVD risk in race

Additional subheadings were added under section 4: “Cardiovascular Risk of Hormone Therapy”. One sub-section is “Mechanisms of CV Risk and Metabolic Factors”. Under this section we describe potential mechanisms of increased CV risk in patients taking ADT. We included additional references to relevant studies describing these mechanisms (namely insulin resistance, blood glucose changes, obesity, metabolic syndrome, hyperlipidemia). Another paragraph in this sub-section details the point made here about how PCa patients commonly suffer from metabolic disorders even before starting hormonal therapies. We briefly discuss this fact and the implications it has for beginning potentially cardiotoxic therapy.

  1. Since mechanistic studies are not possible in human tissue, the authors can discuss the results from the preclinical model (PMID: 36878484; PMID: 27189011, PMID: 26908406).

A paragraph was added under Section 4: “Cardiovascular Risk of Hormone Therapy”. The sub-section is titled “Preclinical Models”. The paragraph includes references to the murine studies the reviewer provided above. We discuss the mechanistic findings in these studies and how they might shed light on pathology occurring in humans on ADT.

Minor comments

  1. Line 142-143 is not necessary

Line 142-143 is deleted.

  1. The section of Race and PCa can be reduced as the focus is not on prostate cancer.

The section “Race and PCa” as written is focused on PCa and potential genetic/biological and social factors which lead to increased risk in minority populations. First paragraph: African American men more likely to get PCa and more likely to die from it. Second paragraph: possible genetic mechanism for increased incidence and mortality due to PCa. Third paragraph: screening tools and how they may contribute to disparities in PCa. Fourth paragraph: SES and access to healthcare as reasons why minority populations experience worse outcomes. We do agree that the final paragraph detailing potential mechanisms of increased CV risk in Latino and Asian populations did not fit well at the end of this section, so we moved it to the end of the “Race and CV Health” section. Here, we think it better serves our message that there are new and key biological factors which predispose patients to cardiovascular disease and that these need to be considered in the setting of starting hormonal therapy.

  1. The table describes abiraterone acetate and enzalutamide should be combined since there was always overlap between the treatments and patient population.

This data has been consolidated to a table entitled “CV Adverse Effects in Patients Receiving Enzalutamide or Abiraterone vs. No ADT”.

Round 2

Reviewer 3 Report (New Reviewer)

I am convinced with the authors revision and response. i do not have further comments

This manuscript is a resubmission of an earlier submission. The following is a list of the peer review reports and author responses from that submission.

Round 1

Reviewer 1 Report

Dear Authors,

I reviewed with interest the paper entitled “Cardiovascular Effects of Androgen Deprivation Therapy and the Influence of Race”.

First, I would strongly congratulate with the authors for their hard work for the present study, which covers an actual and very interesting topic such as summarizing the available research on the cardiovascular risk on men using different forms of ADT (e.g., GnRH agonists and antagonists, surgical ADT, antiandrogens), with a specific focus on racial differences.

I found the present study interesting and well written - no major concerns with language editing or general fluency.

However, before further evaluating the article, I would recommend revision to be taken to improve the quality of the manuscript - below my specific comments:

- The title is clear and descriptive of what authors have explored in their work, yet my suggestion is to include the term “prostate cancer”, as well as the kind of article/analysis that has been carried out (e.g., “review of the literature”).

- The Introduction provides a background which is relevant to the study, yet it could be further improved. Indeed, although used after in the text, main References are lacking in this specific section (just 1 Reference is provided in the Introduction section). For instance, with regards to lines 45-46, due to the possible occurrence of signs of advanced disease - with the subsequent need of further treatments such ad ADT - also in earlier stage, I would suggest adding some more references of relevant studies on that (e.g., DOI: 10.1097/JU.0000000000001313); moreover, with regards to lines 46-47, I would suggest adding a proper Reference, such as European of Urology guidelines on PCa (e.g., from Uroweb). And so on for the rest of the section which is focused on CV aspects and ADT.  

- The aim of the study is stated at the end of the Introduction section, yet I would suggest stating it a bit more precisely (e.g., what have been assessed as outcome and performed analyses)

- Figures are clear and not repetitive, as well as the Results (i.e., different sub-headings for the analyzed topics).

- Overall, the paper results methodologically correct. However, some main issues in the Materials and Methods section are missing and/or not clearly described or in enough detail. In particular, research strategy should be more precisely presented (e.g., how it has been carried out, time frame of the searching, excluded articles, kind of included articles, reasons for exclusions - also a Table on that should be provided); there are precise guidelines on that. Similarly, how search terms have been used should be more precisely described.

-Conclusion is adequately presented, and interpretations are well stated. However, especially in the light of the stated conclusions (i.e., there is a paucy of evidence on this specific topic of interest) some other and main interesting works on the cardiovascular risk associated with these agents could be added in the proper sub-headings within the manuscript (e.g., DOI: 10.3389/fendo.2021.695170). Limitations of the study are properly acknowledged.

I have not further suggestions.

Reviewer 2 Report

The authors submitted a review article on the association between androgen deprivation therapy (ADT) for prostate cancer and cardiovascular events (CV).

The basic description of ADT may seem somewhat verbose to urologists. However, non-urologists may find it useful to deepen their basic understanding of ADT. The explanation of how various types of ADT increase the risk of CV is very easy to understand, with appropriate citations to clinical research papers.

The authors point to socioeconomic factors as one reason for the higher risk of CV among African Americans. They also point to African American distrust of the health care system. As a non-American reviewer, it is difficult for me to fully judge this distrust. Therefore, I would appreciate the opinions of other reviewers.

Reviewer 3 Report

The field of cardio-oncology has gained much attention recently and it is well accepted nowadays that ADT is associated with significant cardiotoxicities. In fact this topic has been reviewed by multiple manuscripts in the literature (Hu et. al ATVB 2020; Challa et.al Current Treat Options Oncol 2021). The authors attempted to explore the implication of racial disparities in the association between ADT and cardiotoxicity, which was something important that could potentially highlight the significance of the manuscript. Unfortunately they only spent a large part of the manuscript describing the association of race with CV events and PCa separately (both topics again have been reviewed extensively) instead of discussing the implication of race on CV outcome in patient receiving ADT. The authors mentioned in line 331 there is "lack of investigation into the CV risk profile of different races while receiving ADT". If that is the case, perhaps a review on the topic should wait until more primary literature is available.